# Effect of Neonicotinoid Exposure on the Life History Traits and Susceptibility to *Plasmodium* Infection on the Major Avian Malaria Vector *Culex pipiens* (*Diptera*: *Culicidae*)

Romain Pigeault [1,2,*], Danaé Bataillard [1], Olivier Glaizot [1,3] and Philippe Christe [1]

1   Department of Ecology and Evolution, University of Lausanne, CH-1015 Lausanne, Switzerland; danae.bataillard@unil.ch (D.B.); olivier.glaizot@unil.ch (O.G.); philippe.christe@unil.ch (P.C.)
2   Equipe EES, Laboratoire EBI, Université de Poitiers, UMR CNRS 7267, F-86073 Poitiers, France
3   Musée Cantonal de Zoologie, CH-1014 Lausanne, Switzerland
*   Correspondence: romain.pigeault@unil.ch

**Abstract:** *Culex pipiens* complexes play an important role in the transmission of a wide range of pathogens that infect humans, including viruses and filarial worms, as well as pathogens of wildlife, such as the avian malaria parasite (*Plasmodium* spp.). Numerous biotic and abiotic stresses influence vector-borne pathogen transmission directly, through changes in vector density, or indirectly by changing vector immunocompetence, lifespan, or reproductive potential. Among these stresses, mosquito exposure to sublethal doses of pesticides could have important consequences. In addition to being exposed to pollutants in aquatic breeding sites, mosquitoes can also be exposed to chemicals as adults through their diet (plant nectar). In this study, we explored the impact of mosquito exposure at the larval and adult stages to one of the most commonly used pesticides, imidacloprid, a chemical belonging to the class of the neonicotinoids, on a set of life history traits ranging from development time to fecundity. We also studied the impact of this pesticide on the susceptibility of mosquitoes to infection by the avian malaria parasite, *Plasmodium relictum*. Surprisingly, we observed no effects of imidacloprid on any of the parameters examined. This result highlights the fact that *Culex pipiens* mosquitoes do not appear to be susceptible to imidacloprid when exposure doses are close to those measured in the field.

**Keywords:** imidacloprid; pesticide; vectors; sub-lethal dose; field-realistic dose; fecundity; survival; development time

## 1. Introduction

Over the past 50 years, the world's human population has more than doubled. This drastic increase has led to a change in land use with, in particular, strong agricultural expansion [1,2] associated with the development of intensive agriculture and an increased use of chemical inputs (fertilizers, phytosanitary products [3]). Although the benefits of these inputs have been significant [4,5], since the beginning of the 21st century, an increasing number of countries have introduced laws to reduce their use. Many phytosanitary products have indeed been associated with major health and environmental problems [6–8]. However, for economic reasons, some countries seem to be backtracking, and chemicals that were under a moratorium are now back on the market. For instance, 18 European countries (out of 28) voted for a new five-year authorization of glyphosate in 2017 [9], and the French Parliament authorized the temporary return of neonicotinoids in 2020 to "save" the beet industry.

The main controversy over the use of pesticides stems in particular from their non-targeted effects [10–12]. There is growing evidence that pesticide use at lethal and sublethal concentrations contributes to the decline of many invertebrate species that provide ecosystem services [13], such as pollination [14,15] and crop pest predation [16–18]. Although

much less publicized, many non-target species, living in ecosystems very different from those associated with the crops in which the pesticides are sprayed, may also be impacted [19,20]. Indeed, some pesticides, particularly neonicotinoids, have high persistence and high water solubility, causing them to accumulate in aquatic ecosystems. They are frequently detected in ground and surface waters, including rivers, lakes, and waterholes [21,22]. Organisms living in these environments may be exposed throughout their lives to sublethal concentrations of these pollutants, which may ultimately impact their life history traits. Reductions in survival, growth, and reproduction of freshwater organisms, particularly aquatic invertebrate species, can alter ecosystem functions related to decomposition and nutrient cycling [13,20].

The presence of sublethal concentrations of pesticides in aquatic systems, particularly in ponds and wetlands, from nearby agricultural activities, may have important consequences for the transmission dynamics of vector-borne diseases. Natural and artificial water bodies near agricultural land are productive breeding sites for many species of mosquitoes [23–25]. The impact of pesticide exposure may, as observed in several invertebrate species, impact development, notably by disrupting the midgut structure [26], immunity [27–29], and survivorship [10,26] of mosquitoes, which could ultimately modify their vectorial capacity. Moreover, the sublethal concentration of pesticides found in aquatic environments can select for insecticide resistance in invertebrate vectors [30–32], and therefore negatively impact vector control strategies.

In addition to being exposed to pesticides in breeding sites, mosquitoes can also be exposed to chemicals as adults through their diet. Although female mosquitoes are blood-feeders, sugar sources, such as nectar, fruits, and phloem sap compose an important part of their diet [33,34]. Pesticides found in plant nectar [35,36] have been described as having a negative impact on the life history traits and immunocompetence of several herbivorous invertebrate species [37–39].

In this study, we explored the impact of the exposure of mosquitoes, at the larval and/or adult stage, to one of the most commonly used pesticides, imidacloprid, on a set of life history traits ranging from development time to fecundity. We also studied the impact of this pesticide on the susceptibility of mosquitoes to infection by the malaria parasite, *Plasmodium*. Imidacloprid is a systemic insecticide that acts as an insect neurotoxin, and is a representative neonicotinoid. This molecule has high water solubility and a relatively long half-life, and therefore has the potential to accumulate in soils and leach to surface water and groundwater [22]. The concentrations of imidacloprid in the aquatic ecosystems have ranged from 0.001 to 320.000 µg/L [22]. In addition, this pesticide is also detected in the nectar and pollen of flowering crops, typically at concentrations ranging from 0.7 to 10.0 µg/Kg [40]. The concentrations of imidacloprid used in this study were selected in order to expose mosquitoes to field-realistic doses of imidacloprid corresponding to what can be found in suitable aquatic systems to the development of mosquito larvae (3 µg/L, Supplementary Materials Table S1) and in the nectar of plants (0.8 µg/Kg [41]) for adult exposure. Experiments were conducted with a natural system consisting of the avian malaria parasite *Plasmodium relictum* and its vector in the wild, the mosquito *Culex pipiens* [42].

## 2. Results

In this experiment, the mosquitoes were assigned to one of four different treatment groups (160 larvae per group): (1) both larvae and adults emerged from this group were unexposed to imidacloprid (control), (2) only adults emerged from this group were exposed to imidacloprid, (3) only larvae from this group were exposed to imidacloprid, and (4) both larvae and adults from this group were exposed to imidacloprid.

Imidacloprid exposure did not have significant effect on larval mortality (Table 1, Figure 1), development time (Table 1, Figure 2A), or adult mosquito size (Table 1, Figure 2B). Males had a shorter development time (Table 1, Figure 2A) and were smaller than females (Table 1, Figure 2B).

**Table 1.** Impact of *Culex pipiens* exposure to imidacloprid on a set of life history traits, ranging from development time to susceptibility to avian malaria parasite infection. Resp. variable: response variable, Expl. Variable: Explanatory variables, Mean: mean of the parameter, SE: standard error, IC: confidence interval, Test: likelihood ratio test or *F* test, Model nb: the different statistical models built to analyse the data are described in the Supplementary Materials Table S1. Unexposed and unexp.: unexposed to imidacloprid, Exposed and exp.: exposed to imidacloprid.

| Resp. Variable | Expl. Variable | Mean ± SE or IC | | Test | *p* | Model Nb. |
|---|---|---|---|---|---|---|
| Survival rate | Larvae exposure status | Unexposed<br>Exposed | 0.71, IC: 0.66–0.76<br>0.76, IC: 0.71–0.80 | $\chi^2_1 = 1.54$ | 0.215 | 1 |
| Development time | Larvae exposure status | Unexposed | 10.17 days ± 0.11 | $F = 0.160$ | 0.688 | 2 |
| | | Exposed | 10.76 days ± 0.10 | | | |
| | Sex | Female<br>Male | 11.33 days ± 0.07<br>9.06 days ± 0.07 | $F = 461.4$ | **<0.001** | |
| | | Exposure status: Sex | | $F = 1.230$ | 0.268 | |
| Wing size | Larvae exposure status | Unexposed<br>Exposed | 0.302 cm ± 0.02<br>0.310 cm ± 0.03 | $F = 0.050$ | 0.817 | 3 |
| | Sex | Female<br>Male | 0.337 cm ± 0.02<br>0.280 cm ± 0.01 | $F = 591.8$ | **<0.001** | |
| | | Exposure status: Sex | | $F = 3.020$ | 0.083 | |
| Blood meal rate | Larvae exposure status | Unexposed<br>Exposed | 0.89, IC: 0.82–0.96<br>0.91, IC: 0.84–0.98 | $\chi^2_1 = 0.13$ | 0.719 | 4 |
| | Adult exposure status | Unexposed<br>Exposed | 0.88, IC: 0.81–0.95<br>0.92, IC: 0.86–0.98 | $\chi^2_1 = 0.63$ | 0.427 | |
| | | Larvae: Adult exposure status<br>Larvae and Adult unexp.<br>Larvae and Adult exp.<br>Larvae exp. and Adult unexp.<br>Larvae un-exp. And Adult exp. | <br>0.84, IC: 0.72–0.96<br>0.89, IC: 0.79–0.99<br>0.92, IC: 0.83–1.00<br>0.94, IC: 0.86–1.00 | $\chi^2_1 = 1.89$ | 0.169 | |
| Blood meal size (haematin, µg) | Larvae exposure status | Unexposed<br>Exposed | 21.49 ± 1.06<br>21.57 ± 0.96 | $\chi^2_1 = 0.11$ | 0.742 | 5 |
| | Adult exposure status | Unexposed<br>Exposed | 21.86 ± 1.01<br>21.17 ± 1.00 | $\chi^2_1 = 0.54$ | 0.463 | |
| | | Larvae: Adult exposure status<br>Larvae and Adult unexp.<br>Larvae and Adult exp.<br>Larvae exp. and Adult unexp.<br>Larvae un-exp. and Adult exp. | <br>21.41 ± 1.34<br>20.86 ± 1.22<br>22.16 ± 1.43<br>21.58 ± 1.71 | $\chi^2_1 = 0.29$ | 0.169 | |

**Table 1.** *Cont.*

| Resp. Variable | Expl. Variable | Mean ± SE or IC | | Test | *p* | Model Nb. |
|---|---|---|---|---|---|---|
| Fecundity (nb. of eggs) | Larvae exposure status | Unexposed<br>Exposed | 211.74 ± 6.76<br>207.39 ± 7.41 | $\chi^2_1 = 0.01$ | 0.975 | 6 |
| | Adult exposure status | Unexposed<br>Exposed | 218.76 ± 7.26<br>197.04 ± 6.11 | $\chi^2_1 = 2.65$ | 0.104 | |
| | | Larvae: Adult exposure status<br>Larvae and Adult unexp.<br>Larvae and Adult exp.<br>Larvae exp. and Adult unexp.<br>Larvae un-exp. and Adult exp. | <br>219.00 ± 9.62<br>194.00 ± 8.76<br>218.55 ± 11<br>201.18 ± 8.41 | $\chi^2_1 = 0.01$ | 0.950 | |
| Parasite burden (nb. of oocysts) | Larvae exposure status | Unexposed<br>Exposed | 61.98 ± 7.84<br>62.55 ± 5.98 | $\chi^2_1 = 0.29$ | 0.587 | 7 |
| | Adult exposure status | Unexposed<br>Exposed | 65.34 ± 7.44<br>59.04 ± 5.99 | $\chi^2_1 = 0.30$ | 0.589 | |
| | | Larvae: Adult exposure status<br>Larvae and Adult unexp.<br>Larvae and Adult exp.<br>Larvae exp. and Adult unexp.<br>Larvae un-exp. and Adult exp. | <br>73.2 ± 13.73<br>65.58 ± 8.64<br>60 ± 8.35<br>52 ± 8.19 | $\chi^2_1 = 2.23$ | 0.135 | |

Bold in the table means: $p < 0.05$.

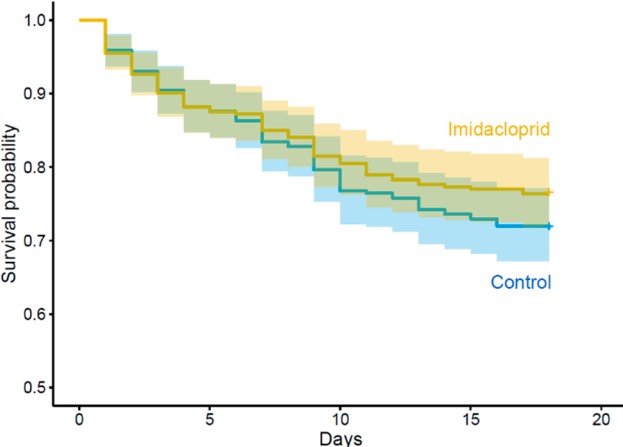

**Figure 1.** Effect of imidacloprid on the survival probability of *Culex pipiens* larvae, shown as Kaplan–Meier survival curves. Blue: larvae unexposed to pesticide (control), yellow: larvae exposed to imidacloprid (concentration: 3 µg/L).

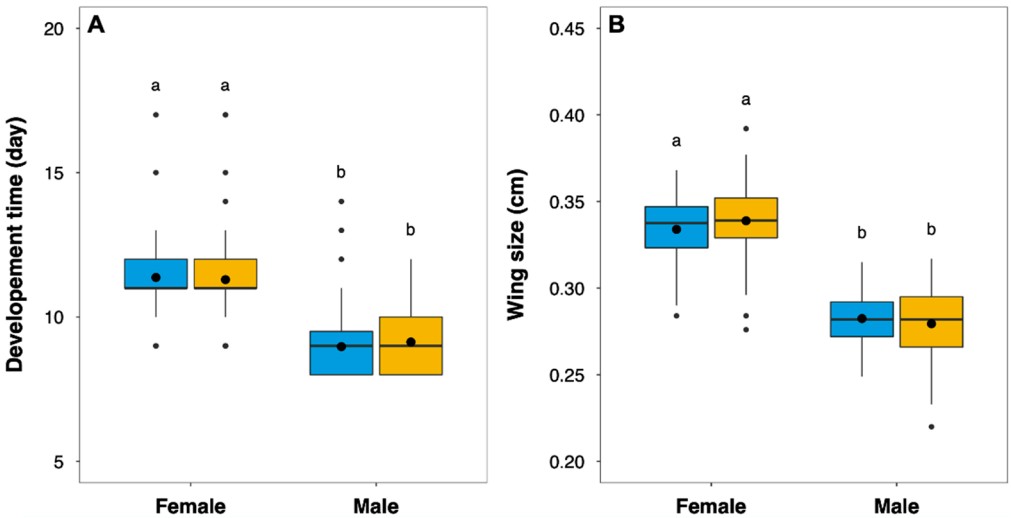

**Figure 2.** Effect of imidacloprid on (**A**) larval development time (day) and (**B**) adult wing size (cm) for both *Culex pipiens* mosquito sexes. Blue: larvae unexposed to pesticide (control), yellow: larvae exposed to imidacloprid (concentration: 3 µg/L). Levels not connected by the same letter ("a"; "b") are significantly different ($p < 0.05$). Boxplots represent the means (points) and medians (horizontal lines). Boxes above and below the medians show the first and third quartiles, respectively. Lines delimit 1.5 times the inter-quartile range, above and below which individual counts are considered outliers and marked as black dot "·".

Imidacloprid exposure at the larval and/or adult stage did not have a significant effect on either the proportion of females that took a blood meal (Table 1) or the amount of blood ingested (Table 1, Figure 3A). No significant effect of imidacloprid exposure was observed on the number of laid eggs (Table 1, Figure 3B). A positive relationship was observed between blood meal size and the number of laid eggs (model 6: $\chi^2_1 = 24.39$, $p < 0.0001$).

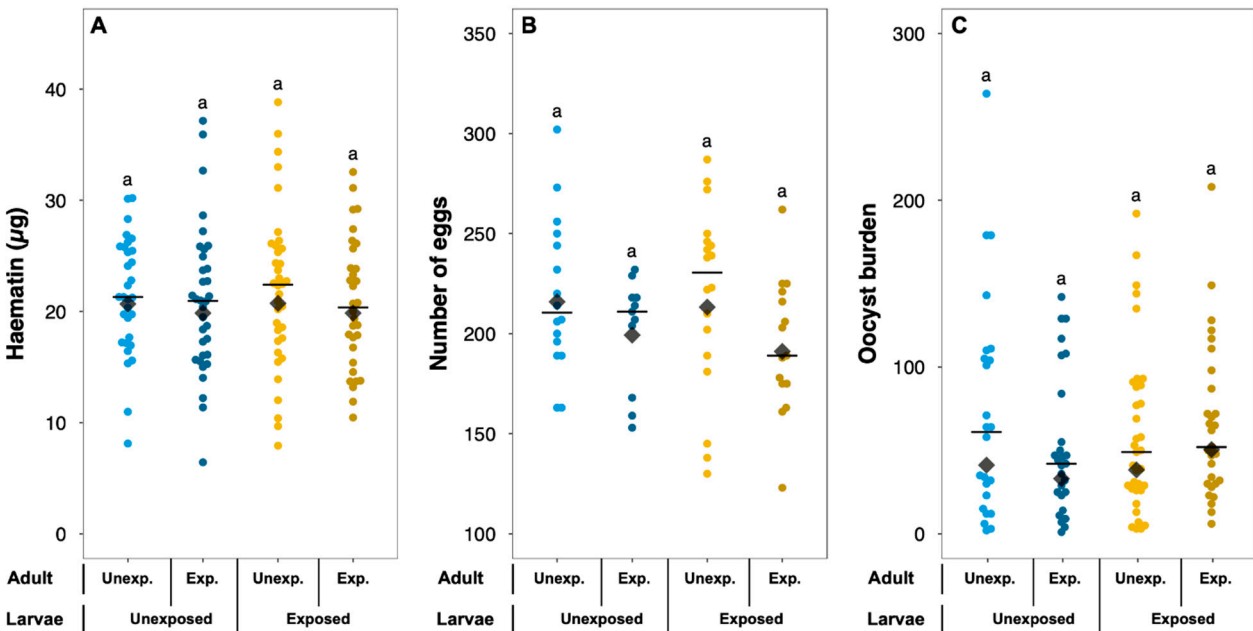

**Figure 3.** Effect of imidacloprid exposure at the larval or adult stage on (**A**) females' blood meal size, (**B**) fecundity (number of eggs laid), and (**C**) susceptibility to avian malaria parasite infection (oocyst burden count in *Culex pipiens* mosquito midgut one week post-blood meal). Each dot corresponds to an individual. Black horizontal lines represent medians, and black diamonds represent means. Blue: larvae unexposed to pesticide, yellow: larvae exposed to imidacloprid (concentration: 3 µg/L). Dark blue and dark yellow: adult females exposed to imidacloprid (concentration: 0.8 µg/Kg). Levels connected by the same letter ("a") are not statistically significantly different ($p > 0.05$).

Midgut dissection revealed that 100% of the mosquitoes fed on infected bird blood were infected with *Plasmodium relictum*. Imidacloprid exposure at the larval and/or adult stage had no effect on the oocyst burden of mosquitoes (Table 1, Figure 3C). A positive relationship was observed between blood meal size and oocyst burden (model 7: $\chi^2_1 = 10.35$, $p < 0.001$).

## 3. Discussion

*Culex pipiens* complex plays an important role in the transmission of a wide range of pathogens that infect humans, including viruses and filarial worms, as well as pathogens of wildlife, such as the avian malaria parasite (*Plasmodium* spp. [42–44]). Numerous biotic and abiotic stresses can impact the life history traits and the immunocompetence of these vectors, and ultimately impact the transmission of pathogens [45–49]. Here we assessed the consequences of larval and/or adult exposure to imidacloprid, one of the most widely used pesticides worldwide, on *Culex pipiens* life history traits and susceptibility to avian malaria parasite infection. We observed no effects of this pesticide on any of the parameters examined in this study. Based on the analysis of survival rates of *Culex pipiens* larvae 72 h after exposure to different doses of imidaclopird, we demonstrated in a previous study that the mosquito line used in this experiment was not resistant to this pesticide (concentrations of imidaclopird that kills 50% of the test animals ($LC_{50}$): 30 µg/L, see [48]). The fact that no effect was observed in the present study is probably explained by the low concentrations of imidacloprid used (3 µg/L in water and 0.8 µg/Kg in sugar solution). These low concentrations were selected based on a literature review, in order to expose mosquitoes to field-realistic doses of imidacloprid, corresponding to what is found in suitable aquatic systems, for the development of mosquito larvae [22,50,51] and in the nectar of plants for adult exposure [41,52].

A worldwide and European survey of imidacloprid residues in aquatic systems showed an average concentration of less than 1 ng/L [53,54]. Focusing on environments suitable for mosquito larvae development (e.g., wetlands, ditches, water bodies), the

average concentrations of imidacloprid found in Canadian wetlands were less than or equal to 15.9 ng/L [51]. Imidacloprid concentrations measured in water samples collected from flood control ditches adjacent to conventionally sprayed blueberry fields in British Columbia, Canada, ranged from 3.2 to 1459.0 ng/L [55]. Numerous surveys in agricultural regions of California have shown that the maximum concentration of imidacloprid found in surface water ranged between 1.38 and 3.29 µg/L [50]. The concentration used in this study, 3 µg/L, was therefore in the upper range of what is generally reported from the field (Supplementary Materials Table S1). Nevertheless, some studies have found very high imidacloprid concentrations, but these observations may be considered to be quite out of the ordinary [22,50,51]. The highest reported concentrations in aquatic systems was observed in Dutch agricultural surface waters at concentrations up to 320 µg/L [22]. Concerning concentration found in plant nectar, imidacloprid spreads to the nectar and pollen of flowering crops, typically at concentrations ranging from 0.7 to 10.0 µg/kg [40], but a maximum concentration of 0.8 µg/kg was reported in a large study conducted in the United States. This result was considered representative by the European Food Safety Authority [41], and was therefore used in this study.

To date, few studies have explored the impact of imidacloprid on the life history traits of mosquitoes, most of them exposing larvae only to pesticide concentrations higher than what is generally reported in surface waters. By exposing mosquito larvae to concentrations 2 to 1000 times higher than the one used here, studies have highlighted a negative effect of this pesticide on the survival, development, and swimming behavior of the larvae [26,56–59]. Studies have also shown consistent results in other aquatic invertebrate species when using relatively high concentrations of imidacloprid [56,60]. For instance, survival of the stonefly larvae (*Pteronarcys dorsata*) was significantly reduced but only from a concentration of 48 µg/L [61]. Nevertheless, very low concentrations of pesticide can have significant effects on some species. Imidacloprid reduced the survivorship, feeding, and egestion of mayflies (*Epeorus longimanus*) and oligochaetes (*Lumbriculus variegatus*) at concentrations greater than 0.5 but less than 10.0 µg/L [62]. Larval growth of Chironomid (*Chironomus tentans*) was significantly reduced when the concentration of imidacloprid in the water exceeded 3.5 µg/L. The length of harlequin fly larvae (*Chironomus riparius*) is negatively impacted by imidacloprid from very low concentrations of pesticide (less than 1 ug/L [63]), and this pollutant also reduced the emergence of the harlequin fly by 10% at 2.09 and 50% at 3.11 µg/L during a 28-day toxicity test (see discussion in [63]). *Culex pipiens* mosquito larvae therefore appear, like the majority of aquatic invertebrate species studied, to be vulnerable to imidacloprid [26,59,64], but only at concentrations higher than those generally reported in the field. The sensitivity of aquatic invertebrate species to imidacloprid is therefore highly variable, and this heterogeneity may be due to various factors associated with differences in the adsorption of chemical substances by the exoskeleton, respiratory strategy, or by the body size and shape [65]. In addition, it has also been suggested that inter- as well as intraspecies differences in oxidative detoxification capacity could be a mechanism that may explain variation in the susceptibility of individuals to imidacloprid [66,67].

With regards to adult exposure, no studies have been conducted to date on the effect of the ingestion of imidacloprid on the life history traits of mosquitoes. Our experimental design has allowed us to demonstrate that the ingestion of imidacloprid by females, whether or not it is associated with larval exposure, had no impact on the probability of having a blood meal, on the blood meal size, fecundity, or susceptibility to avian malaria parasite infection. We suspected a possible effect of this pesticide on the susceptibility of mosquitoes to infection by ingested pathogens, because negative effects of imidacloprid on the immune system of invertebrates [27,37,38,68–70], as well as on their microbiota [71], have been reported in several species. Immunity and the intestinal microbiota of mosquitoes have been shown to play a key role in their susceptibility to *Plasmodium* infection [72–76]. However, most of these studies again revealed negative effects of imidacloprid at concentrations higher than those generally found in the field [37,68–70,77]. An effect of imidacloprid on

the susceptibility of mosquitoes to *Plasmodium* infection has also been suspected, since this pesticide appears to alter the midgut of mosquitoes. One study showed that exposure to low-dose imidacloprid during the larval stage disrupts the development of the digestive tract of *Aedes aegypti* [26]. The midgut epithelial membrane is the first barrier that the *Plasmodium* parasite must cross to infect their vector. The damage of this organ could lead to a less efficient physical barrier favoring infection. Here, we did not study the midgut structure of *Culex pipiens*, but we did not find any difference in the ability of the mosquitoes to digest blood (the amount of haematin produced was similar between the four groups of mosquitoes), nor in the parasite load in the midgut wall one week after the bloodmeal.

## 4. Conclusions

In conclusion, we did not find any effect of the neonicotinoid imidacloprid on the life history traits of *Culex pipiens* mosquito and on its susceptibility to infection by avian malaria parasite. We deliberately used low concentrations of the pesticide, in order to come as close as possible to natural conditions. However, aquatic organisms are generally exposed to a multitude of pollutants in their living environment [55]. It would be relevant to study all possible outcomes of these stressor combinations (i.e., additive, synergistic, or antagonistic) on mosquito life history traits. In addition, an important limitation of our experimental design is that all female mosquitoes were exposed to infected birds and developed parasites. This makes it impossible to independently explore the effect of exposure to each of these two stressors. Another limitation is that the mosquitoes were exposed to imidacloprid over a single generation. Exposure to sublethal doses of pesticide may not impact the life history traits of individuals, but may have a negative impact on their offspring [78,79]. For instance, the exposition of the ladybird *Coccinella septempunctata* and the heteroptera *Orius sauteri* to sub-lethal doses of imidacloprid negatively impact the life history traits of the progeny of the directly exposed individuals [80,81]. Future studies should be conducted to investigate the transgenerational effects of exposure to field-realistic doses of neonicotinoide on mosquito life history traits. It is also important to mention that in this study, both mosquito larvae and adults were reared under optimal conditions (the larvae were reared individually, food ad libitum, optimal temperature and humidity). Food availability is known to permit mosquitoes to cope with the detrimental effect of *Plasmodium* infection [82]. It is thus not possible to rule out that these rearing conditions mask a potential effect of exposure to low concentrations of imidacloprid. A recent study conducted on the same biological system showed that exposure to a low dose of glyphosate associated with nutritional stress tends to increase the probability of being infected by avian malaria parasite [48].

## 5. Materials and Methods

### 5.1. Malaria Parasites and Bird Infection

*Plasmodium relictum* is one of the most widespread avian malaria parasites in western Europe [83]. The *Plasmodium* strain used in this study comes from infected great tits (*Parus major*) captured in Lausanne (Switzerland) in April 2019. *P. relictum* (lineage SGS1) was identified from a blood sample using molecular methods. For this purpose, a nested PCR [84] was performed on the blood sample after DNA was extracted using a DNeasy Blood and Tissue Kit (Qiagen, Hombrechtikon, Switzerland), according to the manufacturer's instructions. Nested PCR products were sequenced as described in Rooyen et al. [85], and identified by performing a local BLAST search in the MalAvi database (http://mbio-serv2.mbioekol.lu.se/Malavi/ (accessed on 23 April 2019) [86]). The parasite was then injected intraperitoneally (i.p.; [87]) into uninfected canaries (*Serinus canaria*), and then maintained in the laboratory until the experiment (August 2019) by making four passages i.p.

Five canaries were used to carry out this experiment. Prior to the experimental infection, a small amount (ca. 3–5 µL) of blood was collected from the metatarsal vein of each bird to ensure that they were free from any previous malaria and malaria-like

infections [83]. Birds were then inoculated by i.p. injection of 100 μL of an infected blood solution composed of phosphate-buffered saline (PBS) and blood (1:1 ratio) sampled from two canaries infected with the parasite three weeks before the experiment. Ten days post-infection, the percentage of infected red blood cell (parasitaemia) of each bird was estimated by visual quantification of blood smears [83]. The three birds with the most similar parasitaemia were selected and used to feed mosquitoes (parasitaemia of the three selected birds measured the day before the experiment: bird 1 = 0.24%, bird 2 = 0.49%, bird 3 = 0.59%). All individuals were treated at the end of the experiment to clear the infection with a dose of commercially available Malarone.

### 5.2. Mosquito Rearing and Imidacloprid Exposure

The experiment was carried out with *Culex pipiens* mosquitoes, the principal vector of *Plasmodium relictum* in our study population (Lausanne, Switzerland) [42]. Mosquitoes were collected in the field and maintained in insectary under standard conditions (26 ± 1 °C, 70% ± 5% RH, and 12 h light/12 h dark photoperiod) since August 2017. In a previous study, we showed that the mosquito line used here was not resistant to imidacloprid (LC50: 30 μg/L; see [48]). Mosquito eggs used in this study were obtained by feeding 50 females on one healthy bird (*Serinus canaria*). Eggs were then placed in a plastic tank filled with mineral water. Freshly hatched larvae (5–10 h old) were then randomly collected and assigned to one of four different treatment groups (160 larvae per group): (1) both larvae and adults emerged were unexposed to imidacloprid (control), (2) only adults emerged were exposed to imidacloprid, (3) only larvae were exposed to imidacloprid, and (4) both larvae and adults were exposed to imidacloprid. Larvae were reared individually in plastic tubes (30 mL) containing 6 mL of mineral water or 6 mL of imidacloprid solution (concentration 3 μg/L, imidacloprid dissolved in mineral water, 98% purity; Sigma-Aldrich, Switzerland). The imidacloprid concentration used here was chosen after calculating the geometric mean of the maximum values measured in suitable aquatic environments for mosquito larvae development (see Supplementary Materials Table S1). Mineral water and imidaclopride solution were changed every 3 days to ensure that the mosquito larvae were exposed to the same concentration of imidaclopride throughout their growth. The larvae were fed daily with 0.5 mg of food (1:1 rabbit pellets and fish food). Mortality and larval development time were monitored on a daily basis until adult emergence. Straight after emergence, all mosquitoes belonging to the same treatment were moved in the same adult rearing cage. Mosquitoes were fed ad libitum on either a 10% glucose solution or 10% glucose imidacloprid solution (imidacloprid concentration: 0.8 μg/Kg), according to their treatment. The concentrations of imidacloprid used in sugar solution was chosen based on the concentration of imidacloprid found in nectar of oilseed rape [41], as well as in previous studies on the impact of imidacloprid on invertebrate species [40,88].

Eleven days after the birds were infected (see above), three feeding cages were prepared, each containing one acutely infected bird (cage 1: bird 1, cage 2: bird 2, cage 3: bird 3). To minimize defensive behavior, the canaries were immobilized in plastic tubes, keeping only their legs accessible to the mosquitoes (for more details, see [89]). At 18 h 00, 12 ± 2 female mosquitoes (5 ± 2 days old) from each treatment were placed together inside each feeding cage. To distinguish between the treatments, females were previously marked with fluorescent powders of different color (the color assigned to each treatment changed between the three cages; for more details, see [90]). At 21 h 00, all blood-fed females were removed from the cages, briefly anaesthetized with $CO^2$, counted and put individually into 30 mL plastic tubes covered with a net. A piece of cotton impregnated with 10% glucose water solution or 10% imidacloprid glucose solution (0.8 μg/Kg) was placed on top of each tube according to mosquito treatment. Five days after the blood meal, all females were placed in new plastic tubes filled with 4 mL of mineral water to stimulate them to lay eggs. The amount of haematin present in each first tube was quantified as an estimate of the size of the blood meal [90]. Three days later, (day 8 post-blood meal), females were anaesthetized with $CO^2$, and one wing was removed from each female and

measured under a binocular microscope along its longest axis as an index of body size [91]. All females were then dissected to count the number of *Plasmodium* parasite (oocyst stage) present in their midguts. The egg-rafts were collected, and the number of eggs was counted under a binocular microscope. The wing size was also measured for all males.

### 5.3. Statistical Analysis

All statistical analyses were performed on R (version 3.4.1) on RStudio v1.3.1056. The sample sizes included in each analysis and the statistical models used to analyze the data are described in the Supplementary Materials Tables S2 and S3, respectively. Larval survival rate was analyzed using Cox proportional hazards regression model (coxph, survival package). The explanatory variable was imidacloprid exposure. A generalized linear model with normal distribution of errors was used to test for difference in development time (calculated as the number of days from hatching to emergence) and adult size (measured as wing length, mm) among larval treatments. Explanatory variables were imidacloprid exposure and mosquito sex. Blood meal rate (proportion of females which took a blood meal), blood meal size (µg), number of eggs, and oocyst burden were analyzed, fitting bird as a random factor into the models, using lmer or glmer (package: lme4 [92]) according to whether the errors were normally (blood meal size, number of eggs, and oocyst burden) or binomially (proportion of females which took a blood meal, prevalence) distributed. Imidacloprid exposure at the larval and adult stage was used as a fixed factor. Blood meal size was also added as a fixed factor when it was not a response variable.

Maximal models were simplified by sequentially dropping non-significant interactions and explanatory variables to establish a minimal model [93]. The significance of the explanatory variables was established using either a likelihood ratio test or an *F* test [93]. The significant Chi-square or *F* values given in the text are for the minimal model, whereas non-significant values correspond to those obtained before the deletion of the explanatory variable from the model. Where appropriate, contrast analyses were carried out by aggregating factor levels together and by testing the fit of the simplified model, using a likelihood ratio test or an *F* test [93].

### 5.4. Ethical Statements

All the authors were trained in animal experimentation by the Lemanic Animal Facility Network (RESAL). The infection of the birds was carried out at the University of Lausanne. The infection protocol, as well as the mosquito feeding protocol, were approved and authorized by the Swiss Federal Office of Food Safety and Veterinary Medicine (permit number 1730.4, date of approval: 16.03.2017, authorization renewed on 16.07.2020). Protocol has been designed to minimize stress and suffering to individuals, as well as the use of the smallest possible number of birds, as required by the 3Rs principle (Replacement, Reduction and Refinement).

**Supplementary Materials:** The following are available online at https://www.mdpi.com/2673-6772/1/1/3/s1, Table S1: Studies used to calculate the geometric mean of the maximum concentration of imidacloprid measured in aquatic environments favorable to the development of mosquito larvae, Table S2: Sample size per mosquito group, Table S3: Description of statistical models used in the study. File S1: Data_Imidacloprid_Pigeault.

**Author Contributions:** R.P., D.B., O.G. and P.C. conceived and planned the experiments. D.B. carried out the experiments, and R.P. carried out analyses. R.P., D.B., O.G. and P.C. contributed to the interpretation of the results. R.P. wrote the initial draft, with modification by O.G. and P.C. All authors helped shape the research, analyses, and interpretation. All authors have read and agreed to the published version of the manuscript.

**Funding:** This research was funded by the Swiss National Science Foundation, grant number 31003A_179378.

**Institutional Review Board Statement:** The study was conducted according to the guidelines of the Swiss Federal Council of Food Safety and Veterinary Office and approved by the cantonal Ethics Committee Vaud (authorization number 1730.4, date of approval: 16 March 2017).

**Informed Consent Statement:** Not applicable.

**Data Availability Statement:** The data presented in this study are available in the Supplementary Materials: File S1: "Data_Imidacloprid_Pigeault_xls".

**Conflicts of Interest:** The authors declare no conflict of interest.

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
