# Peer review of "Effect of Neonicotinoid Exposure on the Life History Traits and Susceptibility to Plasmodium Infection on the Major Avian Malaria Vector Culex pipiens (Diptera: Culicidae)"

_parasitologia, doi:10.3390/parasitologia1010003_

Round 1

Reviewer 1 Report

Comments on the article entitled “Effect of neonicotinoid exposure on the life history traits and susceptibility to Plasmodium infection on the major avian malaria vector Culex pipiens (Diptera: Culicidae)” by Romain Pigeault and coauthors. This is a really nice experiment on the effects of an insecticide exposure on the survival and suitability to parasite infections (among other variables) by mosquitoes. I always enjoy reading the papers by this group.

After reading the paper, I have the following suggestions:

  • I recommend authors to introduce some information in the introduction regarding the insecticide used here. Most of the information in this section is not supported by references and readers need to finish the paper to understand the different levels of insecticides used in other countries.
  • I have checked the reference supporting the concentration of insecticide used in larvae without any success. Using this data, I obtained values lower than those reported here (3 μg/L). Please, clarify this issue in your manuscript (e.g. add this data in supplementary material).
  • The order of exposure of mosquitoes to infected birds may affect the parasite load of birds, according to a previous study of the authors. I strongly recommend to consider this variable in the analyses to check any potential difference in the level of infection of birds exposed to mosquitoes of the different treatments. Are there any effect of the intensity of infection on the parasite load of mosquitoes? Mosquitoes (3 cages per treatment) of the different treatments were exposed to the same (3) birds? Are there any effect of the experimental treatment on the survival of mosquitoes exposed / unexposed to infected bloodmeals?
  • Authors discussed their results based mainly on the low dose used in this study. However, there are many different possibilities which could also explain their results including, for example, nutrition of mosquitoes which allow them to minimize the cost of insecticide exposure.
  • A potential limitation of this study is the fact that all birds were exposed to infected birds and developed parasites. This avoid authors to consider the join effect of the exposure to both stressors (insecticide and parasites) and their cost. This fact could be discussed in the paper.
  • Add the sample size of each level. Add days (?) to the x-axis of fig. 1.

I hope that all these comments allow authors to improve the quality of this nice study.

Have a nice day and stay safe!

Author Response

We are very grateful for the time Reviewer 1 has spent evaluating our work and for his constructive suggestions which greatly improved our manuscript.

Please find below (in bold) our responses to the reviewer’s comments.

Comments on the article entitled “Effect of neonicotinoid exposure on the life history traits and susceptibility to Plasmodium infection on the major avian malaria vector Culex pipiens (Diptera: Culicidae)” by Romain Pigeault and coauthors. This is a really nice experiment on the effects of an insecticide exposure on the survival and suitability to parasite infections (among other variables) by mosquitoes. I always enjoy reading the papers by this group.

After reading the paper, I have the following suggestions:

  • I recommend authors to introduce some information in the introduction regarding the insecticide used here. Most of the information in this section is not supported by references and readers need to finish the paper to understand the different levels of insecticides used in other countries.

R1-1: Thank you for this comment. We agree and added information about imidacloprid in the introduction (lines 76-81).

  • I have checked the reference supporting the concentration of insecticide used in larvae without any success. Using this data, I obtained values lower than those reported here (3 μg/L). Please, clarify this issue in your manuscript (e.g. add this data in supplementary material).

R1-2: We added the table S1 published by Morrissey et al 2015 in our supplementary material (Table S1) where we highlighted the studies used to calculate the geometric mean of the maximum concentration of imidacloprid measured in aquatic environments favorable to the development of mosquito larvae (we excluded estuaries, rivers, groundwater, streams and lakes).

  • The order of exposure of mosquitoes to infected birds may affect the parasite load of birds, according to a previous study of the authors. I strongly recommend to consider this variable in the analyses to check any potential difference in the level of infection of birds exposed to mosquitoes of the different treatments.

R1-3: We are not sure to understand this comment. In our last study (Isaïa et al 2020, 10.1098/rspb.2020.2615) we showed that the parasite burden within mosquitoes is impacted by biting order, however no effect was observed on the parasite load within the vertebrate host (the parasite load in the birds did not change between the beginning and the end of the experiment). In the imidacloprid experiment, as in the vast majority of our studies published so far (except in Isaïa et al 2020), we unfortunately did not follow the order of mosquito bites, we collected all the blood fed females at the end of the experiment.   

  • Are there any effect of the intensity of infection on the parasite load of mosquitoes?

R1-4: Indeed, we observed a significant effect of bird (F=16.75, p < 0.001). Here is the parasitaemia (percentage of infected red blood cells) measured in each of the three birds and the respective average oocyst burden in mosquitoes: bird 1: 0.24%, 62 ± 3.77; bird 2: 0.49%, 70 ± 9.09; bird 3: 0.59%, 68.01 ± 6.62. Mosquitoes fed on the least infected bird have a lower oocyte burden (post hoc test, bird 1/ bird 2 = p < 0.0001, bird 1/ bird 3 = p < 0.001, bird 2/ bird 3 = 0.845). But as the difference in the transmission rate of the parasite from bird to mosquito is not interesting in its own right here, it was controlled by leaving bird ID as a random factor in the model.

  • Mosquitoes (3 cages per treatment) of the different treatments were exposed to the same (3) birds? Are there any effect of the experimental treatment on the survival of mosquitoes exposed / unexposed to infected bloodmeals?

R1-5: First we have prepared three cages each containing one infected bird. Then, in each cage, we released 12 ± 2 females from each experimental treatment at the same time (to distinguish between the treatments, mosquitoes were previously marked using different fluorescent colour powders). All the birds were thus exposed to approximately the same numbers of mosquitoes from each treatment.

We dissected all the blood fed females which did not allow us to monitor the survival rate of the adult females. During the week between the blood meal and the day of dissection, 18 females died and there appeared to be no effect of treatment on the mortality rate (4 females died in two treatments and 5 died in the other two). Unfortunately, we did not have monitored the survival rate of males.

  • Authors discussed their results based mainly on the low dose used in this study. However, there are many different possibilities which could also explain their results including, for example, nutrition of mosquitoes which allow them to minimize the cost of insecticide exposure.

R1-6: We totally agree. We now acknowledge this potential bias at the end of the discussion (lines 231-239).

  • A potential limitation of this study is the fact that all birds were exposed to infected birds and developed parasites. This avoid authors to consider the join effect of the exposure to both stressors (insecticide and parasites) and their cost. This fact could be discussed in the paper.

R1-7: Thank you for this comment. We agree and now mention this point in the discussion (lines 221-224).

  • Add the sample size of each level. Add days (?) to the x-axis of fig. 1.

R1-8: We added a supplementary table with the sample size of each mosquito group (table S2). We also change the x-axis label of the figure 1.

I hope that all these comments allow authors to improve the quality of this nice study.

R1-9: Thank you for these pertinent comments and suggestions.

Have a nice day and stay safe!

Reviewer 2 Report

Wonderful paper, concise and easy to read. The authors have performed elegant experiments, exposing Culex mosquitoes to relevant concentrations of the insecticide imidacloprid (a neonicotinoid). The authors found absolutely no difference in immature development and survival, adult wing size, bloodmeal rate and size and malaria infection success between the treated and control groups.  

I only have a few minor comments to improve the clarity of the paper:

Discussion 144-159: Can you relate those values with the values you used, as I had to go back and forth between this paragraph and the previous one to remind myself about the experimental concentrations?

Discussion 160-179: Can you add possible explanations for those differences in observations, rather than reporting concentrations and effects only?

Methods 243-245: Can you add the susceptibility data here? I was wondering about their resistance profile from the beginning, which I found at the end (discussion).

Methods 258-259: Every 3 days to ensure exposure to the same concentration. Why 3 days, is there evidence that 3 days is the threshold?

Methods 270: What does ‘dpi’ stand for?

Methods 270-272: Can you briefly describe how the birds were immobilized?

Author Response

Please find below (in bold) our responses to the reviewer’s comments.

Wonderful paper, concise and easy to read. The authors have performed elegant experiments, exposing Culex mosquitoes to relevant concentrations of the insecticide imidacloprid (a neonicotinoid). The authors found absolutely no difference in immature development and survival, adult wing size, bloodmeal rate and size and malaria infection success between the treated and control groups.  

We would like to thank Reviewer 2 for very useful comments.

I only have a few minor comments to improve the clarity of the paper:

  • Discussion 144-159: Can you relate those values with the values you used, as I had to go back and forth between this paragraph and the previous one to remind myself about the experimental concentrations?

R2-1: Done, thank you.

  • Discussion 160-179: Can you add possible explanations for those differences in observations, rather than reporting concentrations and effects only?

R2-2: We agree, we added possible explanations for those interspecific variations in susceptibility to imidacloprid (see lines 186-192).

  • Methods 243-245: Can you add the susceptibility data here? I was wondering about their resistance profile from the beginning, which I found at the end (discussion).

R2-3:  We now indicate lines 267-269 that the mosquito line used in this experiment is susceptible to imidacloprid.

  • Methods 258-259: Every 3 days to ensure exposure to the same concentration. Why 3 days, is there evidence that 3 days is the threshold?

R2-4: Imidacloprid is rather stable when diluted in water (Mahapatra et al 2017). However, the light accelerates the degradation of this molecule (Mahapatra et al 2017). Moreover, photodegradation of imidacloprid in water is also influenced by temperatures (Liu et al 2015). According to the results published by Mahapatra et al 2017 (see figure 2) and in view of the rearing conditions of the mosquito larvae (26 ± 1°C, 70 ± 5% RH and 12L: 12D photoperiod, water PH = 7.7) we decided to renew all the solutions (control and imidacloprid) every 3 days to ensure that the larvae are always exposed to a concentration of imidacloprid close to 3 µg/L.

Mahapatra et al 2017: 10.1007/s00128-017-2159-6

Liu et al 2015: 10.1080/03601230600701775

  • Methods 270: What does ‘dpi’ stand for?

R2-5: dpi stands for “days post infection” (as defined line 235). But as we used this acronym only twice we now used "days post infection" instead of "dpi".

  • Methods 270-272: Can you briefly describe how the birds were immobilized?

R2-6: We now briefly describe how we immobilized the birds (see lines 293-297).

Reviewer 3 Report

Author edits seem sufficient 

Author Response

We are very grateful for the time Reviewer 3 has spent evaluating our work and for his constructive suggestions/corrections which greatly improved the first version of our manuscript.

This manuscript is a resubmission of an earlier submission. The following is a list of the peer review reports and author responses from that submission.

Round 1

Reviewer 1 Report

Here, the authors exposed larvae and/or adult Culex pipiens mosquitoes to field-relevant dosages of imidaclorprid and assessed life history traits, including blood feeding and Plasmodium infection. The authors found no effect of imidaclorprid on life history traits of this mosquito species. Overall the manuscript is well written, with data neatly presented. My one major concern is that the authors make large claims concerning the effects of this insecticide on Culex. The authors completed exposure experiments to one generation of Culex mosquitoes and saw no immediate effect on life history traits. While I agree with their findings, the authors need to acknowledge the limits to this experimental design in the discussion, as changes to insect life history traits may occurs after exposure of multiple generations to sub lethal doses of insecticides. I look forward to following the work of these authors and hope they explore the potential effects of these field-relevant dosages on multiple generations of mosquitoes in future experiments.

Reviewer 2 Report

The manuscript "Effect of neonicotinoid exposure on the life history traits and susceptibility to Plasmodium infection on the major avian malaria vector Culex pipiens (Diptera: Culicidae)" is a laboratory based study trying to test whether exposure to a pesticide affected the life history traits of mosquitoes from the Culex pipiens complex, and their ability to  get infected with Plasmodium relictum, an avian malaria parasite.

Comments

1) A major flaw of the study is that mosquitoes were fed on live animals and nowhere is presented the IACUC Animal use protocol permit, not fulfilling the ethical guidelines of this journal (and of MDPI, more generally):

https://www.mdpi.com/journal/parasitologia/instructions#rethics

2) A second major flaw was the experimental infection of mosquitoes using infected birds, for which again there is no experimental description about the bird infection protocol nor any ethical clearance related to the experimental infection of birds.  Similarly, parasitaemias in the host birds were not measured and any potential heterogeneity related to this was not accounted in the experimental design (This is in no way solved by just analyzing birds as a random "categorical" or "intercept" factor  in the mixed effects models, since that does not account for similar parasitaemias. In this sense, it is worth checking the work by Sinden et al at the Imperial College). It is surprising the infections reached a 100% of the exposed mosquitoes (which seems to imply an even more surprising results of 100% feeding ... ), and that mean number of Oocysts was relatively low in light of what classical studies reported. For example, see:

https://www.jstor.org/stable/3274055?seq=1

Which, by the way, was done at a time when ethical standards were different.

3) The development times from hatching to adults seem relatively longer than what meta-analyses have reported for the Culex pipiens complex

https://academic.oup.com/ee/article-abstract/42/4/614/366992

4) The statistical analysis is presented in a highly mystified way. Especially since normal GLMs and GLMERs were used it is really surprising why authors chose to go for likelihood ratio tests to test significance when a simpler Z wald test is more than a robust approximation to the asymptotic distribution of the parameters in the normal models: 

https://www.google.com/books/edition/Regression/EQxU9iJtipAC?hl=en&gbpv=1&pg=PA663

Also, the writing is high disorganized as it is not stated what test (z or chisq) is used for which distribution.

5) The selection of the concentration for the neocotinoid is not well documented. In this sense the design of the experiment is extremely poor, as ideally range of concentration should have been tested. It is not clear why estimate from the field measurement were taking at face value, considering only the mean and disregarding how variable were the measurements generating that mean.

6) It seems that the manuscript language was not carefully revised. For example, the abstract starts by saying "Culex mosquito complex" where it should say something like "Culex pipiens complex". On top of those imprecisions there are plenty of grammatical errors were verbal tenses are wrong and plurals are not used correctly and many words are misspelled like "traitment" instead of "treatment" in the supplement or things like writing (Diptera: Culicidae) in italics.  

Reviewer 3 Report

Overall, it was a very well written manuscript. I noticed a few minor corrections that needed to be addressed such as italicizing scientific names. My only other comment would be the results section. I felt the way it was presented made it hard to read and interpret - long sentences with multiple colons - made it confusing to keep things straight. It was hard to follow. I might suggest trying to put results in a table or something similar to make it a little more clear for the reader. 

Line 12: "spp." should not be italicized 

Line 83: I might suggest replacing "this larvae" with "this group" 

Line 84: Suggest rewording as, "(2) only adults emerged from this group were exposed to imidacloprid,"

Line 85: Suggest rewording as, "(3) only larvae from this group were exposed to imidacloprid and (4) both larvae and adults from this group were exposed to imidacloprid."  

Lines 89 and 92: There are two "model 2"'s listed. They appear to be separate models with separate model outputs. If so, they should be separated as such.  

Line 96: Italicize Culex pipiens

Line 100: Add the mosquito species in this figure description. 

Line 115: Italicize Plasmodium relictum

Line 115: Add the mosquito species in this figure description.

Line 140: "spp." should not be italicized

Line 149: Add an "s" at the end of sample - should be plural based on the wording of the sentence. 

Line 191: Remove the "s" at the end of effects or remove the "a" before negative if it is in fact multiple negative effects that are being seen. 

Line 192: Sentence seems incomplete as it ends with "but see". But see what? If you are wanted the reader to "see" what's in references 27 and 38 then that should be specified and specify what they are looking for in those references. 

Line 214: Italicize Parus major

Line 215: Italicize Serinus canaria

Line 216: Italicize Plasmodium

Line 220: Sentence seems incomplete - there needs to be a statement after "as". Perhaps "...as described in Rooyen et al. [77] and..."

Line 233: Italicize Plasmodium relictum

Line 236: Italicize Serinus canaria

Lines 258 and 274: Italicize Plasmodium